# The Human Meconium Metabolome and Its Evolution during the First Days of Life

**DOI:** 10.3390/metabo12050414

**Published:** 2022-05-05

**Authors:** Nihel Bekhti, Florence Castelli, Alain Paris, Blanche Guillon, Christophe Junot, Clémence Moiron, François Fenaille, Karine Adel-Patient

**Affiliations:** 1Département Médicaments et Technologies pour la Santé (DMTS), MetaboHUB, CEA, INRAE, Université Paris-Saclay, 91191 Gif-sur-Yvette, France; bekhtinihel@gmail.com (N.B.); florence.castelli@cea.fr (F.C.); blanche.guillon@cea.fr (B.G.); christophe.junot@cea.fr (C.J.); 2UMR7245 MNHN-CNRS, Muséum National d’Histoire Naturelle, 75005 Paris, France; alain.paris@mnhn.fr; 3Clinique Sainte Thérèse, 75017 Paris, France; pharma.stherese@almaviva-sante.com

**Keywords:** meconium metabolome, untargeted metabolomics, LC-HRMS, day-to-day variations

## Abstract

Meconium represents the first newborn stools, formed from the second month of gestation and excreted in the first days after birth. As an accumulative and inert matrix, it accumulates most of the molecules transferred through the placenta from the mother to the fetus during the last 6 months of pregnancy, and those resulting from the metabolic activities of the fetus. To date, only few studies dealing with meconium metabolomics have been published. In this study, we aimed to provide a comprehensive view of the meconium metabolic composition using 33 samples collected longitudinally from 11 healthy newborns and to analyze its evolution during the first 3 days of life. First, a robust and efficient methodology for metabolite extraction was implemented. Data acquisition was performed using liquid chromatography coupled to high-resolution mass spectrometry (LC-HRMS), using two complementary LC-HRMS conditions. Data preprocessing and treatment were performed using the Workflow4Metabolomics platform and the metabolite annotation was performed using our in-house database by matching accurate masses, retention times, and MS/MS spectra to those of pure standards. We successfully identified up to 229 metabolites at a high confidence level in human meconium, belonging to diverse chemical classes and from different origins. A progressive evolution of the metabolic profile was statistically evidenced, with sugars, amino acids, and some bacteria-derived metabolites being among the most impacted identified compounds. Our implemented analytical workflow allows a unique and comprehensive description of the meconium metabolome, which is related to factors, such as maternal diet and environment.

## 1. Introduction

Meconium, i.e., the first stools of the neonate, starts accumulating in the fetal intestine from the 12th week of gestation and is excreted within the first 24–79 h post-partum [1,2]. Meconium represents an accumulative matrix with a low metabolic activity. It thus provides the longest historical record of fetal exposure but also contains the essential nutrients to shape the future primordial microbiota. Meconium is composed of ~80% water, and, in a decreasing order of abundance, lipids, proteins, and metabolites; it also contains intestinal epithelial cells, neonatal hairs, and minerals [3]. The different substances found in meconium are either produced by the fetus itself or result from trans-placental transfer. The latter substances notably include metabolites derived from the mother’s endogenous and microbiota metabolisms, and from various maternal exogenous factors (diet, medication, and environmental contaminants).

Targeted analyses of meconium have largely been performed to evidence fetal exposure to specific xenobiotics [4,5]. As a representative example, Ostrea et al. compared the pesticide content detected in hair, umbilical cord blood, and meconium collected from 598 infants, and evidenced higher levels of some xenobiotics in meconium due to its accumulative nature [6]. Other mass spectrometry (MS)-based targeted analyses of meconium evidenced signatures of fetal exposure to alcohol [7], tobacco [8], and drugs [9]. Targeted analyses using liquid chromatography coupled to tandem mass spectrometry (LC-MS/MS) and subsequent quantification of 33 bile acids also showed an association between primary and secondary bile acid concentrations in meconium and gestational age [10].

Few studies have dealt with untargeted metabolomics analyses of meconium, and essentially provided lists of metabolites that discriminate different mother/child health outcomes. For instance, using nuclear magnetic resonance (NMR) analysis, Peng et al. identified nine meconium metabolites allowing the diagnosis of gestational diabetes mellitus (GDM) [11]. Thanks to a set of 113 metabolites detected by LC coupled to high-resolution MS (LC-HRMS) in meconium samples, Chen et al. then described associations between GDM and alterations in taurine, pyrimidine, and bile acid metabolic pathways [12]. On the other side, NMR-based metabolomics described 16 water-soluble metabolites (e.g., amino acids, organic acids, and ketone bodies) while sterols (e.g., cholesterol, squalene) and fatty acids were the major lipid classes detected in the organic fraction. Both the concentrations of nine water-soluble metabolites, on the one hand, and the fatty acid concentrations and their unsaturation index, on the other hand, significantly increased with postpartum time [13], reflecting breastfeeding initiation [14]. An analysis of 70 fecal samples from 21 newborns, collected between day 1 and day 30 after birth, allowed identification and/or quantification of 33 metabolites, the concentrations of which changed during the first days of life and which correlated with intestinal bacterial species appearance [15]. In a recent study, Bittinger et al. performed a multi-omics analysis of meconium/fecal samples collected between day 1 and day 7 after birth. Metabolomic analysis was performed by LC-HRMS using a single liquid chromatography condition. In total, 45 metabolites were identified to show different profiles in samples collected after 16 h post-partum, where more bacteria were detected [16]. Wandro et al. performed a non-targeted gas chromatography (GC)-MS analysis of samples collected from day 7 after birth and annotated a total of 224 endogenous metabolites, including amino acids, bile acids, fatty acids, nucleotides, and sugars [17]. The largest mapping of early feces metabolome was very recently provided by Petersen et al. Within this study, a non-targeted LC-HRMS analysis of 100 meconium samples was performed by Metabolon, Inc (Morrisville, NC, USA), which reported the detection of 714 compounds belonging to different metabolic pathways, including predominately complex lipid species (e.g., lysophospholipids, sphingomyelins), fatty acids, amino acids, xenobiotics, vitamins, and cofactors [18].

In the present study, we aimed to provide a comprehensive description of the meconium metabolome, and provide a list of metabolites identified at a high confidence level. We thus implemented an MS-based metabolomics workflow involving two untargeted distinct and complementary LC-HRMS platforms combined with annotation based on an in-house chemical database comprising more than 1200 pure authentic standards analyzed under identical conditions [19,20,21] and confirmed by MS² analysis. Under these conditions, 229 metabolites were confidently annotated. We then analyzed global and annotated metabolome evolution during the first three days of life. 

## 2. Results

### 2.1. Optimization of the Workflow for Metabolome Analysis of Meconium

To obtain the most precise view of the human meconium metabolome, we devised an original sample preparation protocol for obtaining robust metabolic fingerprints. Thus, we first performed preliminary experiments to optimize the different steps of the sample preparation thanks to a pool of meconium samples. The steps considered and the optimized final conditions are provided in Figure 1. As a first step, manual homogenization of freshly collected samples was performed prior to aliquoting to avoid any topographical position bias [22]. Then, freeze-drying was performed prior to metabolic extraction to allow future standardization from the dry weight while avoiding potential biases linked to sample-to-sample variation in the water content. We observed that 74 to 78% of the initial meconium weight was lost upon freeze-drying, which is consistent with the reported median value of water content in human feces (~75%) [23]. In line with previous observations [16], methanol proved to be the most efficient solvent for metabolites extraction. Thus, 10 mg of freeze-dried meconium were suspended in 750 µL of methanol/water mixture (4:1, *v*/*v*). Meconium dissociation and homogenization were performed using a Precellys apparatus (Bertin Technologies, Montigny-le-Bretonneux, France) and tubes preloaded with ceramic beads, which was the most efficient and reproducible dissociation method of those we tested (e.g., compared to a sonication bath or probe). Metabolomic analyses of the resulting extracts were performed using two complementary LC-HRMS platforms involving either a reversed-phase column with MS detection in the positive ionization mode (C18-ESI^+^) or a hydrophilic interaction liquid chromatography column with detection in the negative ionization mode (HILIC-ESI^−^), allowing the analysis of hydrophobic and polar metabolites, respectively. It is important to mention that we have recently demonstrated the efficiency and robustness of such a protocol to analyze the metabolic profiles of adults feces [24]. The reproducibility of the sample preparation was assessed by analyzing 5 analytical replicates that showed an average coefficient of variation (CV) below 15% for all the metabolite features (see below). Of note, this study focused on the analysis of small-molecular-weight metabolites, the detection of complex lipids (such as lysophospholipids or sphingomyelins) or long-chain fatty acids would imply the use of a dedicated lipidomics platform.

### 2.2. Characterization of the Human Meconium Metabolome

#### 2.2.1. Comprehensive View of the Meconium Metabolome

The LC-HRMS-based metabolomic methods and their associated data processing workflow (see Material and Methods) were applied to analyze 33 meconium samples collected at different time points from 11 newborns (Appendix A). The number and time of excretion varied greatly from one newborn to another, with the collection time ranging from 1 to 79 h after birth. From the analyzed samples, we extracted 9274 and 12,397 features in the HILIC-ESI^−^ and C18-ESI^+^ analytical conditions, respectively. Of these, 6843 and 8555 features were found to be analytically relevant, i.e., satisfying our 3 quality evaluation criteria (biological to blank samples intensity ratio, CV between QCs, and correlation within the diluted QC series; see the details in the Material and Methods section). A given metabolite is commonly detected under ESI conditions as a multiplicity of molecular species (i.e., monoisotopic peak, adducts, dimers, and/or fragments), resulting in an overestimation of the real number of unique metabolites in the meconium samples [25]. Overall, 224 metabolic features from the C18-ESI^+^ conditions and 149 from the HILIC-ESI^−^ conditions matched with accurate RT and *m*/*z* values of a pure standard included in our chemical reference database. Additional MS/MS (MS²) analyses were performed to confirm these putative annotations, and we finally identified 197 and 89 metabolites for the HILIC-ESI^−^ and C18-ESI^+^ conditions, respectively. Only 57 metabolites were identified by both methods, further demonstrating the complementarity of the 2 LC-HRMS platforms. When excluding the overlapping metabolites, 229 unique metabolites were finally identified in meconium with at least 2 orthogonal parameters (retention time, *m*/*z* or/and MS/MS spectra) as proposed by the Metabolomics Standards Initiative (MSI) [26]. The corresponding list of metabolites is provided in Appendix A. 

#### 2.2.2. A Map of the Human Meconium Metabolome

The chemical families of the 229 identified metabolites were assigned using the human metabolome database (HMDB) [27]. The meconium showed a rich metabolic composition covering several molecular families, as summarized in Figure 2a. We observed a major representation of amino acids, peptides, and analogues (32%); carbohydrates and carbohydrate conjugates (15%); and nucleosides, nucleotides, and analogues (13%), with these 3 families covering 60% of the whole meconium metabolome. Within the remaining 40%, we detected mainly organic acids and derivatives (11%) and fatty acids conjugates (7%). The high representation of amino acid, peptide, and analogue families is consistent with previous studies [15,16], and this may also be linked to the high representation of amino acids and derivatives in our chemical library.

Within our meconium samples, some microbial metabolites were identified, such as muramic acid, 2-isopropylmalic acid, secondary bile acids (e.g., lithocholic acid (LCA)), and short-chain fatty acids (SCFAs, e.g., isobutyric acid). Other compounds that may result from tryptophan metabolization by bacteria were also detected, particularly indole derivatives (e.g., indoleacetic acid, indoleacrylic acid, indolelactic acid, indole-3-carboxylic acid, and indoxyl sulfate) [29]. 

The 229 identified metabolites were then repositioned into metabolic pathways [28]. A total of 52 enriched metabolic pathways were identified, and the most enriched ones are shown in Figure 2b. They include biosynthesis and degradation of amino acids pathways (arginine, glutamine, histidine, valine, leucine, phenylalanine, tryptophan, …; amino-acyl-tRNA biosynthesis), and pathways related to the metabolism of caffeine (paraxanthine, theobromine, …) or antibiotics (neomycin, kanamycin, …).

### 2.3. Analysis of the Whole Dataset Evidenced Rapid Evolution of the Meconium Metabolome during the First Days of Life

#### 2.3.1. Preliminary Analysis of the Whole Dataset through Non-Supervised PCA and Hierarchical Clustering

A first global analysis of all the analytically relevant features obtained under the C18-ESI^+^ (8555 features) and the HILIC-ESI^−^ (6843 features) analytical conditions was performed using a non-supervised multivariate analysis (principal component analysis, PCA) to obtain a first rough picture of the samples and the data distribution. The first 2 PCA components explained 42% (C18-ESI^+^, Figure 3A) and 45% (HILIC-ESI^−^, Figure 3B) of the total variance. Almost the same outliers were identified in both conditions, in particular the NB08.270 and NB06.2520 samples. These two samples were not excreted by the same neonate and the other samples collected from the same neonates were not identified as outliers. They do not correspond to a specific gender, and do not reflect a particular excretion time (270 and 2520 min). Moreover, these samples were not close in the analytical sequence, excluding a possible analytical bias during the data acquisition. Thus, there is no obvious reason for explaining the particular behavior of these two samples. They may either have a true but unexplained different metabolic composition or were partially contaminated during their collection from the diaper. As only a few samples were available for this study, we decided to keep all the samples. 

Unsupervised PCA highlighted similar structuration of the whole dataset in both analytical conditions (Figure 3). A time-dependent distribution of samples was evidenced, with a non-linear and progressive change in the overall metabolome composition of the meconium (Figure 3). The proximity of samples with close sampling times, and, to a lesser extent, excreted by the same newborn were highlighted using non-supervised hierarchical clustering (not shown). We observed a group of features with a low intensity in the early collection times that increased over time while another group of features showed an opposite trend.

#### 2.3.2. Canonical and Regression Analyses Confirmed a Major Impact of Time on the Meconium Metabolome

To more deeply characterize the inter and intra metabolic variability within the newborns, canonical analyses were performed between the C18-ESI^+^ and HILIC-ESI^−^ whole datasets, on the one hand, and the dummy matrix built from repeatedly collected newborns combined with the true time scale at which feces were sampled, on the other hand. Both canonical analyses showed that the first principal components (PC1) represented 38.4% and 38.8% of the total variance and revealed nearly continuous variation in the metabolome for both datasets with rather limited intra-newborn variance (Figure 4). Interestingly, PC2s, which represent 17.2% and 24.0% of the total variance, respectively, displayed a particular behavior for 4 newborns (NB10 for the C18-ESI^+^ dataset, and NB08, NB09, and NB11 newborns for the HILIC-ESI^−^ dataset).

The statistical models used here, which considered the meconium sampling time, highlighted a significant correlation between PC1 scores and time for both analytical conditions (regressions not shown). Accordingly, sparse PLS (sPLS) regressions were significantly established between the 500 most informative features selected from either the C18-ESI^+^ or HILIC-ESI^−^ datasets, on the one side, and the time variable, on the other side. 

Among the 500 features retained for sPLS regression analysis from either the C18-ESI^+^ or HILIC-ESI^−^ dataset, the 20 variables that correlated most with the collection time were sorted according to the absolute value of the correlation with the Comp[1] variable in the sPLS regression. In parallel, thanks to the sPLS regression model used, VIP values were also calculated (Appendix A). Whatever the dataset considered, the absolute values of the correlation of the first 20 features explaining the regressions were above 0.862 when their VIP values were higher than 6.50 for the C18-ESI^+^ dataset and 2.50 for the HILIC-ESI^−^ dataset. Unfortunately, among these 20 features, none was assigned to metabolites present in our in-house database. Among the 500 features selected, only 2 annotated metabolites were found for the C18-ESI^+^ dataset (glycyl-leucine, xylulose), whereas 9 were found for the HILIC-ESI^−^ dataset (ribose phosphate, N-acetylglycine, etc.) (Appendix A); however, all were sorted in the less correlated feature sets with a rank above 112.

These global regressions analyzed by sPLS were then reinforced by a mixed model analysis between the Comp[1] scores and the time variable considering the newborn factor as a random factor. For both datasets, every newborn-specific linear regression displayed a positive slope (Figure 5A,C), except for NB05 in the HILIC-ESI^−^ dataset (Figure 5C). Moreover, the quality of the statistical models was estimated by a quantile-quantile diagram (or QQ plot) of the estimated residues (Figure 5B,D). All samples, except those collected for NB09, were predicted very well by such modeling.

#### 2.3.3. Modeling of the Whole Datasets and Metabolic Changes Associated with Time-Dependent Variation

Global PLS regression thus illustrated the association between the metabolomic datasets and collection time. Better modeling of the link between these two variables, i.e., Comp[1] and time, was examined considering a log-transformation of the time variable and curvilinear modeling based on polynomials expressed in log(time) with degrees 4 and 3 for the C18-ESI^+^ and HILIC-ESI^−^ datasets, respectively (Figure 6 and Appendix A). For both datasets, the model prediction was superimposed on the loess modeling of the data well (Figure 6). In addition, the statistical parameters summarizing this curvilinear modeling were conveniently optimized, as shown by the high significance of the coefficients assigned to polynomials equal to or above degree 2 (Appendix A). These 2 curves were adjusted with strongly significant alpha risks of 5.2 × 10^−15^ and 2.2 × 10^−16^, respectively. Interestingly, if we analyze the parallel variations of these 2 curves by discarding the simultaneous variation of log(time) and keeping the higher degree of polynomials, that is, 4 and 3, respectively, we can roughly estimate the allometric variation between C18-ESI^+^ Comp[1] and HILIC-ESI^−^ Comp[1] with an exponentiation coefficient equal to 0.75. When we more precisely considered the log-transformation of C18-ESI^+^ and HILIC-ESI^−^ data predicted using polynomials, as indicated in Figure 6, we obtained the following allometric equation:Predicted-C18 ESI^+^ = (Predicted-HILIC-ESI^−^)^0.7285^ − 0.1034(1)
with adjusted R^2^ = 0.9668, F_1,29_ = 874, and *p*-value < 2 × 10^−16^ (Appendix A), that is, with an exponentiation coefficient of 0.7285, which is very close to 0.75, the roughly estimated exponent indicated above.

### 2.4. Identification of Metabolites Showing Time-Dependent Variations: Complementarity of Non-Supervised and Supervised Analyses

Based on all these results, and to interpret more deeply and directly the observed differences and/or similarities between the samples, we finally performed non-supervised and supervised analyses solely using the 229 annotated metabolites and considering the 33 samples independently of their origin (i.e., not taking into account the fact that stool samples were repeatedly collected for some newborns). We first performed PCA, which globally reproduced the same structuring observed for the whole dataset (Appendix A vs. Figure 3). This suggests that the set of annotated metabolites is quite representative of the whole datasets. Non-supervised hierarchical clustering of the 33 samples based on the 229 annotated metabolites identified 2 clusters of samples (clusters 1 and 2) (Appendix A). The first cluster of samples is composed of 23 meconium samples, of which 20 were excreted before 24 h, while the second cluster is composed of 10 samples, 8 of which were excreted after 24 h. This spontaneous classification suggests a 24-h cutoff regarding the global metabolic composition in our cohort of meconium samples. The intensities of some metabolites increased (mainly enriched in the cluster > 24 h) while others decreased (mainly enriched in the cluster < 24 h) over time (see below). As shown by non-supervised analysis of the whole dataset, evolution of the meconium composition is thus mainly driven by the time post-partum, independently of the newborn. The distinction between samples excreted in the first 24 h from others was already detectable in the PCA score plot (Appendix A).

According to these results, we finally performed univariate and multivariate supervised analyses on the 229 annotated features, with samples classified into 2 groups, i.e., excreted before or after 24 h. First, we performed univariate Wilcoxon testing, using a false discovery rate (FDR) correction for multiple comparisons. Around 70% of the annotated metabolites showed comparable intensities before and after 24 h, whereas 67 out the 229 features (29%) showed significantly different intensities between the 2 groups (adjusted *p*-values < 0.05). Among these metabolites, 50 showed significantly higher intensities in late samples (Appendix A, red color), including N-acetylglycine, L-threonic acid, or methylguanine, but also some potential microbiota-derived metabolites (e.g., indole derivatives, isobutyric acid). The 17 remaining metabolites displayed significantly more intense signals in samples collected before 24 h (Appendix A, blue colors), such as glycerol phosphate, xanthopterin, and N-acetylneuraminic acid. Metabolite up- or downregulation were in agreement with the correlation (negative or positive) calculated in sPLS regression (Appendix A). We finally performed supervised multivariate analysis on the 229 annotated features, (i.e., PLS-DA modeling), considering 2 groups of samples (before or after 24 h). A model was successfully built, with a good predictive value (pR2Y < 0.05, pQ2 < 0.05). We then identified 36 metabolites with variable importance in projection (VIP) values above 1.5, i.e., the metabolites that participate most in the model building and then discriminate the most between the 2 groups of samples (Appendix A, yellow color). Within these discriminant metabolites, some with the highest VIP values were already contributing highly in the t1-component of the non-supervised PCA (Appendix A; e.g., glyceric acid, N-acetyl- glutamine, N-acetyl- aspartic acid, threitol, methyl-succinic acid), and all these metabolites were identified as being impacted by the collection time in the univariate analyses. 

## 3. Discussion

The goal of this study was to provide the most precise view of the meconium metabolic composition, with an emphasis on small-molecular-weight metabolites and excluding complex lipids (such as lysophospholipids or sphingomyelins) or other long-chain fatty acids whose detection would imply the use of a dedicated lipidomics platform. In that context, we first implemented a sample preparation approach to robustly extract metabolites from meconium. This protocol was then applied to 33 samples collected from 11 healthy newborns during their first 3 days of life. Metabolite profiling was then performed using two complementary untargeted LC-HRMS approaches, i.e., reversed phase and HILIC chromatography coupled to a Q-Exactive mass spectrometer. Over 17,000 metabolite features comprising redundancy were identified in meconium samples. On the highest MSI level 1, 216 metabolites were identified thanks to our in-house database and using pure authentic standards with an accurate mass, retention time, and MS/MS matching. An additional set of 13 metabolites were annotated with the same procedure but at level 2 since some isomers could not be distinguished. Overall, 229 metabolites were annotated in meconium, which compares favorably with the 222 unique metabolites counted by Aristizabal-Henao et al. using 3 distinct MS-based platforms (2 LC-HRMS and 1 GC-MS) [30]. Using the same analytical approach and database, we identified more than 400 unique metabolites in adult stools [31], highlighting the comparatively lower metabolite richness of meconium. Of note, only 74 metabolites from our dataset were also described by Petersen et al. within their list of 714 detected metabolites and complex lipids (level of confidence not provided) [18].

Non-supervised analysis and modeling of the whole datasets evidenced that time post-partum drastically affected the meconium composition, which evolves rapidly independently of the newborn. Within 79 h post-partum, quantitative and qualitative changes in the metabolome core found in the HILIC-ESI^−^ dataset were more pronounced than in the C18-ESI^+^ dataset, although these changes appeared highly dynamically coordinated. Interestingly, most of the metabolites highly impacted by time were not annotated, i.e., not present in our in-house database. This probably reflects that they mostly correspond to metabolites derived from microbiota metabolism.

Non-supervised analysis, i.e., PCA and hierarchical clustering, of the 229 annotated metabolites further suggested a 24-h cutoff for the metabolic composition of our samples. Supervised univariate statistical analysis, performed on samples collected before or after 24 h, revealed that 67 metabolites were impacted by the collection time (29% of the annotated metabolites). In total, 36 metabolites of those 67 also had a PLS-DA-derived VIP score > 1.5 (Appendix A). These 67 compounds represented different chemical families, i.e., amino acids, carbohydrates, and organic acids. The corresponding enriched pathways included the pentose phosphate pathway, and glycerolipid, taurine and hypotaurine, and ascorbate and aldarate metabolism. Interestingly, 50 metabolites were accumulated in late samples (Appendix A, red color), in line with different studies [13,15,16]. These increased metabolites included N-acetylglycine, threonic acid, and methylguanine, but also taurine and phenylalanine as already described by an NMR analysis of meconium collected between days 1 and 3 by Righetti et al. [2]. Interestingly, some potential microbiota-derived metabolites were also found within this group of accumulating metabolites (indole derivatives, isobutyric acid), in line with progressive microbiota establishment and activity. On the other hand, an additional set of 17 metabolites displayed significantly decreased signals in samples collected after 24 h (Appendix A, blue colors), such as glycerol phosphate, xanthopterin, and N-acetylneuraminic acid.

Different studies have reported similar trends to those we observed, which were associated with multiple processes starting after birth: the establishment of the child’s intestinal microbiota and its associated metabolic functions [15,32], which lead to the consumption of in utero accumulated metabolites to benefit new bacteria-derived metabolites, the initiation of breastfeeding or infant formula intake [13], and also the intensification of the neonate endogenous digestive and metabolic functions.

Altogether, we implemented an analytical workflow and provided a unique and comprehensive description of the meconium metabolome. We evidenced its rapid change over the first days of life. Core metabolome accumulating in utero is related to factors, such as the maternal diet and environment.

## 4. Material and Methods

### 4.1. Study Subjects

Meconium samples were collected in the maternity ward at Sainte-Thérèse Clinic (Paris, France; February 2019). In total, 33 meconium samples were included in the study, obtained within the first 79 h post-partum from 11 anonymized newborns, 2 of which were girls. All babies were born at term by vaginal delivery. 

Meconium was recovered by scraping stained diaper with a sterile disposable spatula, taking care to not touch the nappy surface. The diapers were stored at +4 °C until sample collection, which was performed within 12 h (median 4 h). Samples were then immediately stored at −20 °C until transport to the laboratory where they were stored at −80 °C. The meconium samples were collected until a change in color and/or texture was noticed, reflecting the appearance of the first stools. To limit experimental bias related to contamination by chemicals, we provided the same diapers to all participants. The newborn code (NBx), gender, and time of sample collection in minutes postpartum (and the equivalent in hours) are provided in Appendix A.

### 4.2. Meconium Sample Preparation

Frozen meconium was further freeze-dried using a Triad^TM^ Labconco (Missouri, USA) freeze dryer with temperatures fixed at 4 °C for the tray and −83 °C for the trap; the vacuum was fixed at 0.180 mbar. Freeze-dried samples were homogenized, aliquoted, and stored at −80 °C until analysis. To precipitate proteins, 10 mg of freeze-dried meconium were suspended in 750 µL of methanol/H_2_O (4:1, *v*/*v*). The samples were then homogenized using a Precellys 24^®^ (Bertin Technologies, Montigny-le-Bretonneux, France) and CK14 ceramic beads (6500× *g*; 4 °C; 3 × 30 s), and then incubated on ice for 1.5 h. After centrifugation (20,000× *g*; 4 °C; 15 min), supernatants (containing metabolites) were recovered and dried under a stream of nitrogen at 30 °C using a TurboVap^®^ concentration workstation (Biotage, France). Samples were stored dried at −80 °C until further analysis. 

The pellets were resuspended in a volume of 800 µL of ammonium carbonate (10 mM pH 10.5)/acetonitrile (ACN) (40:60, *v*/*v*) or H_2_O + 0.1% formic acid (FA)/ACN + 0.1% FA (95:5, *v*/*v*) for chromatographic separation using HILIC and C18 columns, respectively. Quality control samples (QC) were prepared by pooling equivalent volumes of all samples. Dilution series of QC samples were prepared (1/2, 1/4 and 1/8) to allow data filtration. In total, 100 µL of each biological, QC, and diluted QC samples were spiked with 5 µL of a standard mixture (Appendix A). 

### 4.3. Metabolic Profiling

Metabolic profiling experiments were performed by LC-HRMS following optimized protocols routinely used in our laboratory [20,21]. LC-HRMS was performed on an Ultimate 3000 chromatographic system coupled to a Q-Exactive mass spectrometer (both from Thermo Fisher Scientific, Courtaboeuf, France) fitted with an electrospray (ESI) source operating in the positive (ESI^+^) and negative (ESI^−^) ionization modes. LC was performed using two types of columns to obtain a more comprehensive description of the metabolic landscape: C18 (Hypersil GOLD C18 column, 1.9 μm, 2.1 × 150 mm, Thermo Fisher Scientific; ESI^+^) and ZIC-pHILIC (Hydrophilic Interaction Liquid Chromatography; Sequant ZICpHILIC column, 5 μm, 2.1 × 150 mm, Merck, Darmstadt, Germany; ESI^−^). Diluted QC samples were analyzed in triplicates at the beginning of the sequence while non-diluted QC samples were introduced every 5 biological samples for data normalization/standardization purposes.

Raw data (.raw files) were manually inspected using the Qual-browser module of Xcalibur (version 4.1, Thermo Fisher Scientific) and then converted to .mzXML format using MSconvert (ProteoWizard). Peak extraction, peak picking, alignment, and integration were performed using the Workflow4Metabolomics (W4M) platform [32]. Data were filtered based on three criteria: (i) ratio of chromatographic peak areas obtained for+ biological to blank samples > 3, (ii) coefficient of variation (CV) of metabolites in the QC samples < 30%, and (iii) correlation between QC dilution factors and areas of chromatographic peaks > 70%. The output files were used for metabolite annotation and further statistical analyses. Metabolite annotation was performed thanks to an in-house chemical database by matching accurate measured masses and chromatographic retention times to those of more than 1200 pure authentic standards analyzed under identical conditions [19,20,21]. Retention time (RT) tolerances accepted were ±15 and ±90 s for the C18 and ZIC-pHILIC columns, respectively. The mass to charge (*m*/*z*) tolerance was 10 ppm for both the positive and negative ionization modes. Each annotated peak was manually checked on the Qual-browser module of Xcalibur by considering the peak shape, isotope pattern, and presence of the considered peak in at least 6 successive MS scans. To limit the presence of irrelevant peaks, an intensity cut-off of 10,000 and 30,000 was applied for the C18-ESI^+^ and HILIC-ESI^−^ conditions, respectively. 

MS/MS analysis was then conducted on the relevant signals to confirm their annotation. In total, 4 normalized collision energies (NCEs) were used to obtain the optimal MS/MS spectra (10, 20, 40, and 80%). Of note, some isomeric metabolites could not be resolved using the LC-HRMS approach. Thus, some chromatographic peaks could correspond to more than one metabolite (e.g., hexoses).

Statistical analysis of log-transformed data was conducted using the W4M platform version 3.4.4 [32]. Univariate Wilcoxon tests were conducted to compare data, and adjusted *p*-values were calculated taking into account multiple testing (false discovery rate, FDR). Adjusted *p*-values < 0.05 were considered significant. Multivariate principal component analysis (PCA) and partial least squares-discriminant analysis (PLS-DA) were carried out using log-transformed data. Hierarchical classification of the samples and features (centered and reduced data) was also carried out, and represented in the form of a “heatmap”. Complementary statistical analyses, such as sparse partial least squares (sPLS) regression and canonical correlation analysis, were performed on R 3.6.4 (R Core Team 2019 [33,34]). Mixed model analysis was performed using the R package lme4 [35]. Metabolic pathways enrichment analysis was conducted using the KEGG database and the “pathway enrichment” tool of MetaboAnalyst 5.0 [28].

## Figures and Tables

**Figure 1 metabolites-12-00414-f001:**
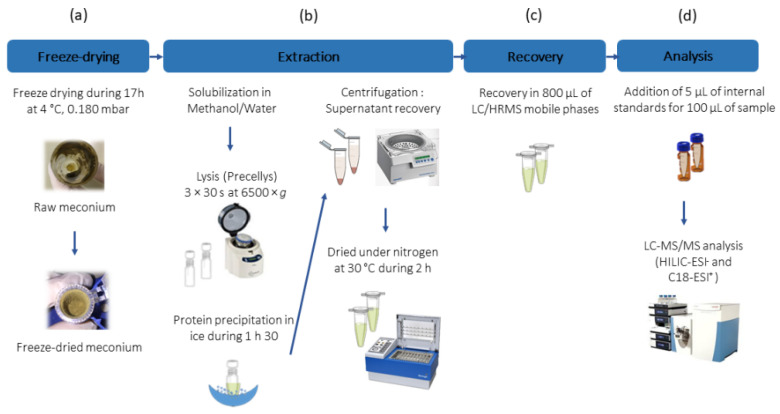
Optimized protocol for meconium preparation prior to LC-HRMS analysis. Four major steps were considered, and the corresponding optimized conditions are provided: (**a**) freeze-drying parameters, (**b**) metabolites extraction method, (**c**) recovery of the pellet containing the metabolites, and (**d**) LC-HRMS analytical conditions.

**Figure 2 metabolites-12-00414-f002:**
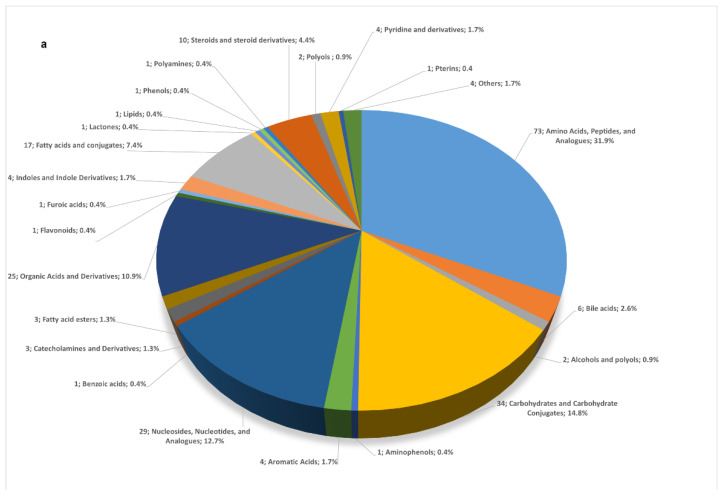
(**a**) Main chemical families represented in meconium and their numbers and percentages (number; name of the chemical family; percentage). (**b**) Enriched metabolic pathways identified through interrogation of the KEGG human metabolic pathways using Metaboanalyst 5.0 tools [28]. The enrichment ratio on the x-axis represents the number of metabolites assigned per metabolic pathway out of the total number of metabolites belonging to the pathway studied.

**Figure 3 metabolites-12-00414-f003:**
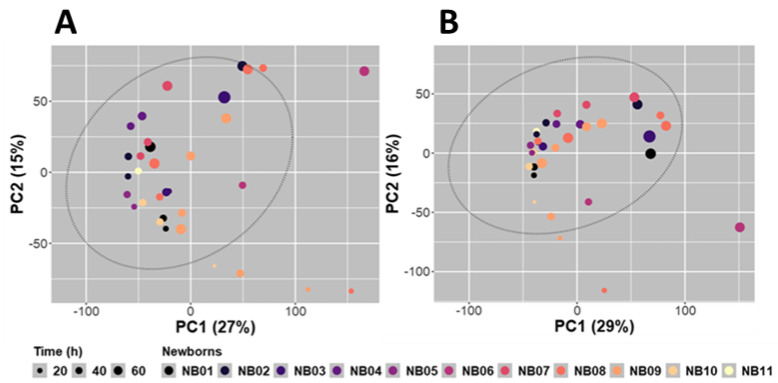
PCA scores plot (PC1 vs. PC2) built using all relevant features obtained in C18-ESI^+^ ((**A**) 8555 features) and HILIC-ESI^−^ ((**B**) 6843 features) MS detection conditions. The colors used represent the different newborns (NB01 to NB11) the meconium samples were collected from. The size of each point traduces the collection time-points, expressed in hours (h). Data were log10-transformed and mean-centered before PCA. Ellipses represent the confidence intervals of the scores projected on factorial plans at a probability *p* = 0.975.

**Figure 4 metabolites-12-00414-f004:**
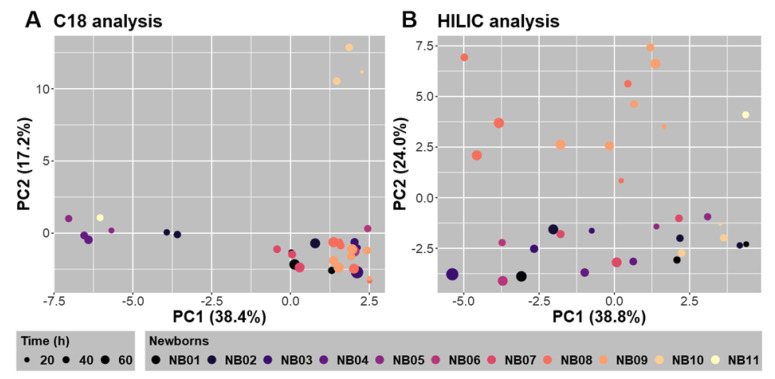
Canonical analyses between all relevant features obtained in C18-ESI^+^ ((**A**) 8555 features) and HILIC-ESI^−^ ((**B**) 6843 features) LC-HRMS conditions, and a dummy matrix summarizing feces samples collected for the different newborns (NBs) completed with the time date (expressed in hours, h) of sample collection.

**Figure 5 metabolites-12-00414-f005:**
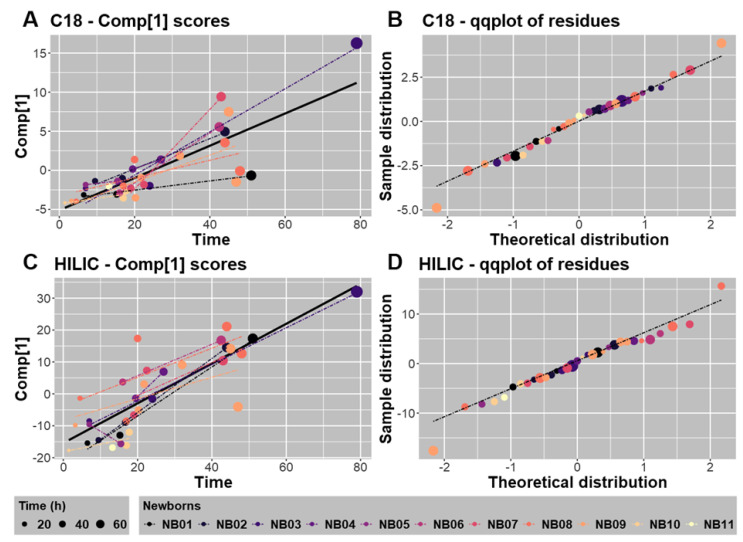
Modeling of the variation in the Comp[1] scores with time (h) for the C18-ESI^+^ (**A**) or HILIC-ESI^−^ (**C**) analytical conditions. Comp[1] scores were calculated according to a sPLS regression between the metabolomic datasets and the time at which fecal matrices were collected. The global regressions were drawn according to the thick continuous black line. For every newborn, a mixed model linear regression was applied to model the regression between Comp[1] scores and time and individual regressions are plotted as dashed lines. The distributions of residues calculated after regression were compared to the theoretical one for metabolites detected under the C18-ESI^+^ (**B**) or HILIC-ESI^−^ (**D**) analytical conditions. The Comp[1] scores were weakly predicted by the mixed model linear regression only for 1 or 2 samples from NB09, both collected after 40 h, in the C18-ESI^+^ (**B**) or HILIC-ESI^−^ (**D**) analytical conditions, respectively.

**Figure 6 metabolites-12-00414-f006:**
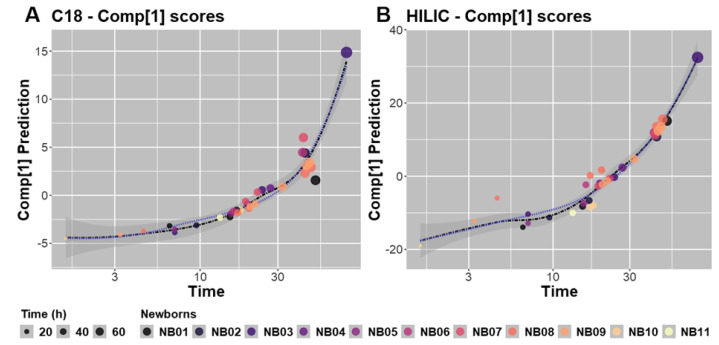
Curvilinear regressions modeled with polynomials of degrees 4 and 3 for the prediction of Comp[1] scores upon log(time) for metabolites analyzed under C18-ESI^+^ (**A**) and HILIC-ESI^−^ (**B**) conditions, respectively. As dashed black lines, the loess-supported regressions are shown with a confidence interval of 5% shown in dark grey, and the dotted blue lines display the polynomial regressions.

## Data Availability

Raw data have been uploaded to MassIVE with the accession number MSV000089260.

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
