# Peer review of "The Human Meconium Metabolome and Its Evolution during the First Days of Life"

_metabolites, 2022, doi:10.3390/metabo12050414_

Round 1
Reviewer 1 Report
Dear authors,
In the work entitled “The human meconium metabolome and its evolution during the first days” of life Bekhti and coworkers describe an innovative methodology to study the metabolome of human meconium during the first days of life. The experimental procedure is well described, and the overall metabolites were listed, however authors show that meconium metabolome change during time without indication of what metabolic changes really occur. In my opinion some data should be clarified.
The title states: The human meconium metabolome and its evolution during the first days of life. Authors prove that there is a change in metabolome however they didn’t state which difference.
In the abstract authors state that: “A progressive evolution of the metabolic profile was statistically evidenced, which probably results both from gut colonization by microbiota, breastfeeding/formula initiation, and increased activity of the neonate digestive and metabolic processes.” and “which would be related to factors such as maternal diet and environment”; since time after birth was the only variable tested this statement is a speculation and in my opinion should not be placed in the abstract.
None of PCA nor sPLS results present the corresponding loadings profile. So, reader didn’t recognize changes of meconium composition upon time. Stated differences are due mainly to which metabolite or metabolite family?
In Figure 2 and Figure 3 respect the order of experiment, changing position may confound readers.
Pag. 7 line 236: What is the meaning of “meconium following a "C" shaped curve”. Please explain how PCA scores shape is relevant.
Pag. 7 line 236: “unsupervised hierarchical clustering further highlighted the proximity of samples with close sampling times, and, to a lesser extent, excreted by the same newborn” may be it would be easier to explain to the reader using hierarchical clustering analysis…
In point 4.2 “Meconium sample preparation” authors used previous to develop this methodology? It was an original research design?
In the conclusions author state that: “time post-partum is the major factor affecting meconium composition” if it was the major factor which other factors were studied?
In the conclusions author state that: “The qualitative and quantitative variations in overall composition of 398 metabolites were highly dynamically coordinated within 79 hours post-partum” please resume the major metabolic changes.
Author Response
We would like to thank the reviewer for his/her comments that allow us clarifying and improving our manuscript.
Below are the answers to all the raised comments (in blue), with all modifications apparent in the “track_changes” version. Accordingly to Assistant Editor’s and Academic Editor’s comments, we also revised the structure order of the manuscript (with separate results and discussion sections) and corrected the references. We also provide a clear version (“_clean”) of the main manuscript to facilitate its reading.

Reviewer 2 Report
Authors described a reliable LC-HRMS method for the screening of metabolites in meconium samples collected from 11 anonymized newborns. The paper is clear and well written. I strongly suggest to accept if in Metabolites.
Author Response
We thank the referee for his/her evaluation of our work and his/her positive comment
Reviewer 3 Report
General Comment: The authors present the paper as a exploratory characterization of metabolites in meconium samples and presumably a perspective on that state of affair. By the concluding statements it is clear they are building to a workflow about the existing and future monitoring and mostly about the LC-HRMS based data collection, QC criteria, pattern recognition-oriented data analysis/interpretation, and database. This is all very well and much needed.
Specific Comment:
- Figure 3: It is not clear whether the authors chose to apply a normalization strategy for PCA analysis? If not, why? Are they confident the same/similar patterns would hold true if the runs were normalized?
- Figure 3: Suggest including corresponding loading plots as well. If possible replace the 2D plots with 3D plots.
- How was extraction efficiency monitored and ensured between samples/batches?
Author Response

(The authors gave the same response as above.)

Round 2
Reviewer 1 Report
Dear authors,
The work has been improved considerably. However there is some problem with the multivariate analysis that needs to be clarified before publication.
Presenting only the PCA or PLS scores does not allow to evaluate the biological significance of the multivariate analysis and explain the discimination of the samples.
If the loadings do not allow you to extract information (the reason for the discrimination) it is because you are not choosing the best graph model to show or overfitting has occurred, caused by too many latent variables used (by the way, what was the number of latent variables you used? it is not mentioned in the article).
Before publication it is necessary to check these points. If you don't have anyone expert in multivariate analysis on your team you should look for external help.
Author Response
We would like to thank the reviewer for his/her additional comments. We apologize as during the first review, we didn’t understand the question – and provided a loading plot for the whole C18-ESI+ dataset (8555 features) as an example.
Indeed, the only way to get some valuable interpretation of multivariate analyses done in the manuscript is to refer to the PCA obtained from assigned features, that this presented in the figure S2A. Combining the PCA (score plot coming from a PCA using 10 components as mentioned now in the figure S2 legend) with the heatmap presented in figure S2B let us to reasonably conclude that there is a clear clustering between samples collected before 24 hours and those collected after 24 hours. We would like to mention that PCA and clustering studies are complementary. We added in figure S2, the plot of loadings corresponding to the component t1 (figure S2C) and to the component t2 (figure S2D, which appears less informative in the time-dependent clustering), these two loadings plot covering only the first 25 highest features entering in the building of these two components. Therefore, from these observations, we can address the time limit at 24 hours to cluster samples – although with some “misclassified” samples. After this first exploration of the dataset, we then try to identify the metabolites discriminating these two subsets of samples, i.e. < 24h versus > 24h. We thus performed univariate and multivariate supervised analysis, through Wilcoxon testing using a False Discovery Rate (FDR) correction for multiple comparisons, and PLS-DA modelling, respectively. All discriminating metabolites identified through the PLS-DA modelling (VIP > 1.5) were shown to be significantly different through univariate analysis (which were already in agreement with the correlation calculated in sPLS regression). Moreover, within these discriminant metabolites, some with the highest VIP values were already highly contributing in the t1-component of non-supervised PCA (figure S2C; e.g. D-glyceric acid, N-acetyl-L-glutamine, N-acetyl-L-aspartic acid, D-threitol, methyl-succinic acid). Altogether, this suggests no overfitting in the modelling. We now mention all these points in the revised manuscript (v2).
We also slightly modified some point in the results and in the discussion (all underlined in yellow in the tracked changes v2 version), notably to further highlight the fact that most of the metabolites impacted by time were not in our in house database. We provide a v2 of the revised manuscript, with all track changes and as a clear version. We hope the revised version of our manuscript will be suitable for publication.
Best regards